# Evaluating the impact of a rapid response system on survival of patients with cancer undergoing emergency surgery for acute abdomen: A single-center retrospective cohort study

Jae Hoon Lee[1], Sang Yun Jung[1], Ki Ho Yu[2], Mee Hee Shin[3], Yun Jung Choi[3], Ra Mi Choi[3], Woo Jin Seo[3], Sang Hee Park[3], Won Ho Han[1]*

1 Critical Care Medicine, National Cancer Center, Goyang, South Korea, 2 Center for Colorectal Cancer, National Cancer Center, Goyang, South Korea, 3 Rapid Response Team, National Cancer Center, Goyang, South Korea

* 13408@ncc.re.kr

## Abstract

Patients with cancer who develop acute abdomen are at high risk of rapid clinical deterioration and often require emergency surgery and intensive care. This retrospective cohort study evaluated the impact of implementing a Rapid Response System (RRS) on survival among 274 patients admitted for emergency surgery at a tertiary cancer center in Korea (145 pre-RRS, 129 post-RRS). Although the post-RRS group had a higher burden of severe illness, including more metastatic disease and higher APACHE II scores, key time intervals from diagnosis to ICU admission, surgery, and antibiotic administration were significantly shorter following RRS implementation. The post-RRS group also demonstrated more favorable postoperative physiological trajectories, including lower postoperative day 7 SOFA scores and greater reductions in SOFA from baseline. Survival to discharge was higher after RRS implementation (85.27% vs. 74.48%, p = 0.039), and RRS activation remained independently associated with improved survival in multivariable analysis (adjusted odds ratio: 3.554, p = 0.021). Patients presenting outside RRS coverage hours had higher severity, longer delays to intervention, and lower survival; in an exploratory counterfactual model, predicted survival increased from 59.4% to 71.0% when RRS availability was hypothetically extended. These findings suggest that RRS implementation may attenuate progression of organ dysfunction and improve survival among high-risk cancer patients with acute abdomen. Expanding RRS operational hours, including continuous 24-hour coverage, may offer additional clinical benefit.

**Data availability statement:** The data underlying this study contain sensitive and potentially identifiable patient information and cannot be made publicly available due to ethical and legal restrictions imposed by the Institutional Review Board of the National Cancer Center, Korea. De-identified data may be made available to qualified researchers upon reasonable request and approval from the Institutional Review Board. Requests for data access should be directed to the IRB office (irb@ncc.re.kr).

**Funding:** Initials of the authors who received each award: WHH Grant numbers awarded to each author: NCC 2310790-3 The full name of each funder: National Cancer Center, Korea URL of each funder website: https://www.ncc.re.kr/indexEn.ncc The funders had no role in study design, data collection and analysis, decision to publish, or preparation of the manuscript.

**Competing interests:** The authors have declared that no competing interests exist.

## Introduction

Each year >18 million new cancer cases are diagnosed worldwide. A significant portion of patients require inpatient care due to complications related to their immunocompromised state. Over half of emergency department visits by cancer patients result in hospitalization, compared to 11.9% for non-cancer patients, highlighting the need for close monitoring in healthcare settings [1,2].

These patients are vulnerable to infections and acute exacerbations, such as acute abdomen, which presents as severe abdominal pain and unstable vital signs, necessitating timely intervention to prevent deterioration [3,4]. Immediate surgical intervention is often required, as delays can result in clinical deterioration. These patients often require intensive care unit (ICU) admission before surgery due to hemodynamic instability and afterward to manage postoperative complications and reduce the risk of mortality, with both phases necessitating advanced support and continuous monitoring [5–7]. Delayed surgical decision-making can impact outcomes. Delayed surgery can lead to increased mortality, longer hospital stays, and higher costs. Patients who experience surgical delays have a 56% higher mortality rate than those who do not, with a 7% longer hospital stay and 6% higher costs. The highest mortality rates are observed in patients undergoing surgery within 6.7–10.7 and 24.5–70.9 hours after diagnosis. Prompt medical intervention and early detection are critical to improving outcomes [8,9].

The Rapid Response System (RRS) plays a vital role in identifying patients at risk of clinical deterioration in general wards. Comprised of ICU-trained physicians and critical care nurses, RRS teams monitor vital signs. When these indicators exceed predefined thresholds, the team intervenes by stabilizing the patient at the bedside or transferring them to the ICU. Studies have demonstrated the effectiveness of the RRS in reducing in-hospital cardiac arrests and mortality rates, primarily in non-cancer patients. One study observed a 49% decrease in out-of-ICU cardiopulmonary arrests, while another reported a reduction in unexpected mortality from 21.9 to 17.4/1,000 discharges after RRS implementation [10,11]. Although studies have evaluated RRS outcomes in patients with cancer, these studies focused on comparing outcomes between cancer and non-cancer groups or were limited to hematologic malignancies [12–14].

To our knowledge, no study has investigated the impact of the RRS on patients with cancer requiring emergency surgery for acute abdomen. Given that these patients are vulnerable to deterioration due to their immunocompromised state, timely intervention is crucial. We address this gap by evaluating the effectiveness of the RRS within this high-risk subgroup and comparing survival rates before and after its implementation in 2019. We sought to identify key prognostic factors to provide insights that may guide the management of patients with cancer in critical condition in general wards.

## Materials and methods

This retrospective single-center study was conducted at the National Cancer Center (NCC), Republic of Korea, and included adult cancer patients (aged ≥18 years) who

developed acute abdomen during hospitalization and required emergency surgery with ICU admission. The study period spanned from March 2016 to August 2023 and was divided into two groups: pre-RRS (March 2016–June 2019) and post-RRS (July 2019–August 2023), based on the implementation of a RRS in July 2019. Acute abdomen was defined as the sudden onset of severe abdominal pain requiring urgent intervention due to intra-abdominal pathology [15]. Exclusion criteria included non-emergency surgery, patients who did not require ICU admission due to mild severity, surgery delayed more than 24 hours after diagnosis, Do-Not-Resuscitate (DNR) orders or Physician Orders for Life-Sustaining Treatment (POLST) status, and cases occurring between 11:00 PM and 6:00 AM, when the RRS was not active.

The RRS operated on weekdays from 6:00 AM to 11:00 PM and consisted of a multidisciplinary team led by two intensivists with expertise in critical care medicine, along with five nurses. The system was activated based on a Modified Early Warning Score (MEWS) of five or higher and specific clinical criteria documented in the real-time medical record, including a respiratory rate below eight or above 30 breaths per minute, oxygen saturation under 85%, heart rate exceeding 140 beats per minute or dropping below 40 beats per minute, systolic blood pressure under 80 mmHg, and arterial blood gas abnormalities, such as a pH below 7.3, $pO_2$ below 55 mmHg, $pCO_2$ above 50 mmHg, or $TCO_2$ below 12 mmol/L. Additionally, activation occurred when nursing records indicated changes in consciousness level, including drowsiness, stupor, or coma, or at the direct request of the attending medical staff [16–18]. Upon activation, a critical care nurse conducted the initial assessment and notified one of the on-duty intensivists, who performed a full clinical evaluation and determined whether ICU transfer or emergency intervention was necessary. During off-hours, patient deterioration was managed by on-call physicians and ICU staff according to standard protocols.

Collected data included demographics (age, sex), cancer type and stage, surgical site and indication, comorbidities (hypertension, diabetes, cardiovascular disease, chronic obstructive pulmonary disease, chronic kidney disease, and liver disease), American Society of Anesthesiologists (ASA) classification, the Acute Physiology and Chronic Health Evaluation II (APACHE II) and the Sequential Organ Failure Assessment (SOFA) scores, the volume of preoperative crystalloid fluid administered, lactate levels, and time intervals from diagnosis to ICU admission, surgery, and initial antibiotic administration. Outcomes included survival to discharge and time to death. Additional clinical interventions such as vasopressor use, mechanical ventilation, and continuous renal replacement therapy (CRRT) were also documented. Organ failure trajectory was assessed using changes in SOFA score from ICU admission to day 7. Because a difference of at least 2 points in the SOFA score represents clinically meaningful organ dysfunction in the Sepsis-3 definition, trajectories were categorized as improvement (ΔSOFA ≤ −2), stability (ΔSOFA −1 to +1), or progression (ΔSOFA ≥ 2). This approach extends established methods of serial SOFA evaluation by applying a validated threshold to distinguish meaningful changes in organ failure status during the early postoperative course [19].

This study was conducted in accordance with the principles of the Declaration of Helsinki and was approved by the Institutional Review Board of the National Cancer Center, Korea (approval no.: NCC 2024−0268). Patients who underwent emergency surgery for acute abdomen between March 2016 and August 2023 were included in this retrospective cohort. Access to the electronic medical records for data verification was from October 14, 2024 to December 14, 2024, following IRB approval. The requirement for informed consent was waived because the study used previously recorded data without direct patient contact.

## Statistical analysis

Baseline characteristics of the pre-RRS and post-RRS groups were compared using the Shapiro–Wilk test to assess distributional normality, followed by the independent t-test or Mann–Whitney U test for continuous variables and the chi-square test for categorical variables. Logistic regression models were constructed with survival to discharge as the dependent variable. Univariable analyses were performed initially, and variables with p values below 0.2 were entered into the multivariable model according to established methodological recommendations [20]. Backward elimination was applied until only covariates with statistical significance at the 0.05 level remained. Because physiologic severity could

be redundantly represented by multiple variables, we examined the relationship between APACHE II, SOFA score, and lactate. Although variance inflation factors did not indicate harmful multicollinearity, APACHE II and SOFA score capture overlapping physiologic domains; therefore, APACHE II was retained as the primary composite severity indicator, while lactate was included as an independent biochemical marker. To account for secular changes in clinical practice over the study period, study year was incorporated as a covariate. Odds ratios with 95% confidence intervals were reported. Kaplan–Meier curves with log-rank testing were used for time-to-event analyses, with surviving patients censored at discharge. Additional subgroup analyses compared perioperative clinical trajectories, including organ support requirements and SOFA score changes, between patients admitted preoperatively and those admitted postoperatively. To assess whether excluding cases presenting outside RRS coverage hours introduced systematic bias, patients evaluated during operational and nonoperational periods were compared separately, with results summarized in S1 Table. Finally, to explore whether the effect of the RRS differed in the most critically ill subgroup, an interaction term between RRS implementation and preoperative ICU admission was incorporated into the multivariable model, and full model results are presented in S2 Table. In a supplementary counterfactual analysis, an adjusted logistic regression model including RRS implementation, nighttime admission, APACHE II score, SOFA score, lactate level, age, and cancer stage was used to estimate the expected probability of survival had patients admitted during noncoverage hours also had access to RRS activation. Counterfactual survival probabilities were generated by fixing nighttime admission and RRS availability while holding other covariates at their mean values. This exploratory analysis was conducted to quantify the potential benefit of extending RRS operation to nighttime hours and does not replace the primary regression findings (S3 Table and S1 Fig.). All analyses were performed using R version 4.4.0(R Foundation for Statistical Computing, Vienna, Austria), and statistical significance was defined by a two-sided p value below 0.05.

## Results

### Participant selection

Among the 424 patients initially assessed for eligibility, 150 were excluded: 62 patients presented with acute abdominal cases between 11:00 PM and 6:00 AM, 52 patients did not require ICU admission due to mild severity, 17 patients experienced delays of over 24 h between diagnosis and surgery, 12 patients opted for DNR orders following diagnosis, and seven patients completed POLST forms. This left a study cohort of 274 patients diagnosed with acute abdomen and admitted to the ICU for emergency surgery, including 145 patients in the pre-RRS period (March 2016 to June 2019) and 129 in the post-RRS period (July 2019 to August 2023) (Fig 1).

### Participant characteristics

Among these 274 patients, 56 died, with septic shock as the most common cause (35 cases), followed by cancer progression (20 cases) and postoperative bleeding (one case). The median age was 69 years, with 45.42% being female. Cancer was localized in 56.03% of patients and metastatic in 43.97%, with perforation (58.76%), bleeding (14.23%), and obstruction (12.41%) as the primary causes for surgery. The median APACHE II score was 25, and the SOFA score was 4. Preoperative fluid administration had a median of 0.30 L, with median times from diagnosis to ICU admission, surgery, and antibiotic administration of 7.12 h, 5.20 h, and 2.23 h, respectively. Vasopressors were used in 28.83% of patients preoperatively and 54.38% postoperatively, with a median duration of 1.53 days. Mechanical ventilation was required in 13.50% of patients preoperatively and 53.65% postoperatively, with a median duration of 2 days. On postoperative day 7, vasopressor use was observed in 5.95% of patients and mechanical ventilation in 9.92%. The median SOFA score on postoperative day 7 was 3, and the median ΔSOFA was −1. Organ dysfunction patterns were categorized as improvement (43.15%), stability (35.08%), or progression (21.77%) based on these serial SOFA assessments. The median ICU length of stay was 1.91 days, and the postoperative hospital stay was 20.64 days. Overall, 79.56% of the patients (218 out of 274) survived to discharge (Table 1).

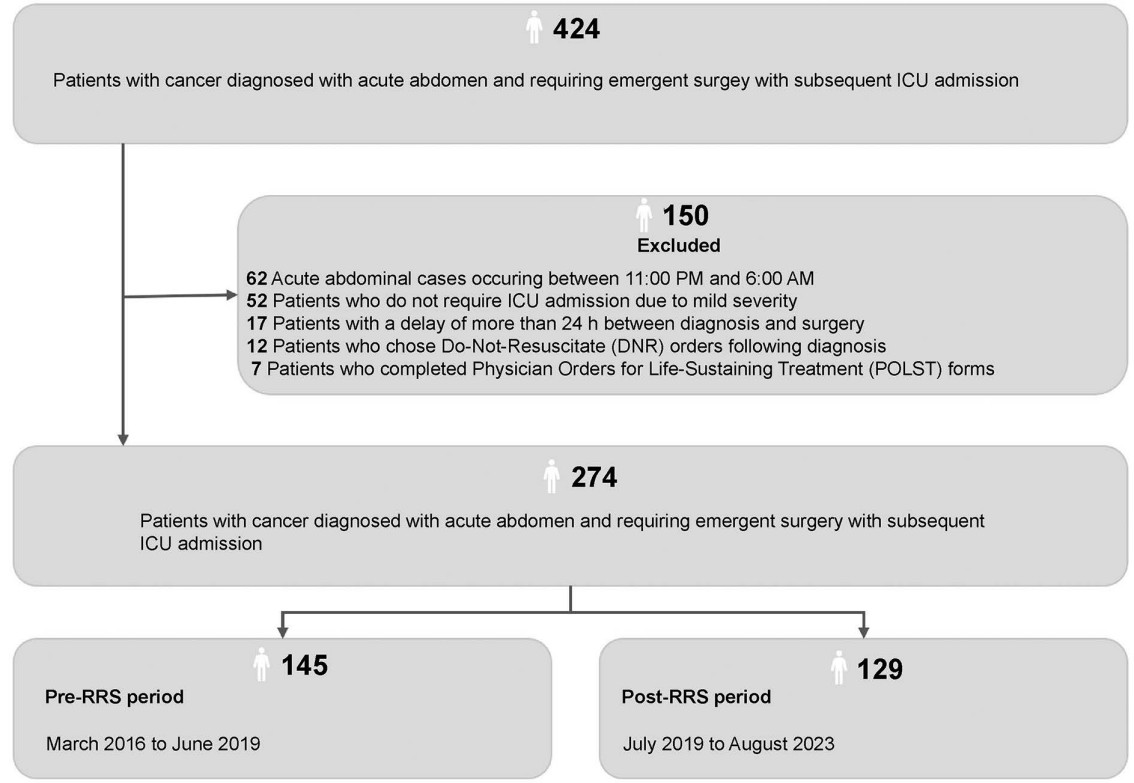

**Fig 1. Eligibility assessment and cohort formation for ICU-admitted acute abdomen patients undergoing emergency surgery (March 2016–August 2023) This figure shows the selection of patients with cancer and acute abdomen who required emergency surgery with subsequent ICU admission.** Of 424 screened patients, 150 were excluded for predefined clinical reasons. The final cohort included 274 eligible patients, comprising 145 in the pre-RRS period and 129 in the post-RRS period.

## Baseline characteristics and clinical outcomes in Pre-RRS and Post-RRS groups

The post-RRS group had a younger median age (68 vs. 72 years, p = 0.041), a higher proportion of metastatic disease (56.03% vs. 31.90%, p < 0.001), and higher APACHE II scores (27 vs. 23, p = 0.009). The volume of preoperative crystalloid fluid administered was greater in the post-RRS group (0.50 L vs. 0.30 L, p = 0.002). Additionally, the time from diagnosis of acute abdomen to ICU admission (5.83 vs. 8.77 h, p < 0.001), surgery (4.57 vs. 6.48 h, p = 0.036), and initiation of antibiotic administration (2.01 vs. 2.50 h, p = 0.042) were shorter. The causes of surgery varied between groups, with perforation (67.44% vs. 51.03%) and bleeding (10.85% vs. 17.24%) as the main indications in the post- and pre-RRS periods, respectively (p = 0.005). CKD prevalence was higher in the post-RRS group (3.10% vs. 0%, p = 0.048). Vasopressor use and mechanical ventilation were higher in the post-RRS group preoperatively (36.43% vs. 22.07%, p = 0.013 and 20.93% vs. 6.90%, p = 0.001, respectively), but postoperative vasopressor use was not significantly different. SOFA scores on the day of operation were similar between groups, whereas SOFA scores on postoperative day 7 (3 vs. 4, p = 0.007) and ΔSOFA values (−2 vs. 0, p < 0.001) differed. Organ dysfunction categories (improvement, stability, progression) also differed significantly between groups (p = 0.001). Other variables, including sex, comorbidities (e.g., hypertension, diabetes), lactate levels, and CRRT use, did not differ significantly. ICU and postoperative hospital stays were similar, but survival to discharge was higher in the post-RRS group (85.27% vs. 74.48%, p = 0.039) (Table 1).

To assess whether excluding patients who presented outside RRS coverage hours introduced systematic differences, we compared patients who deteriorated during RRS coverage hours (N = 274) with those who presented outside coverage

**Table 1. Baseline characteristics and clinical outcomes of ICU patients with acute abdomen.**

| | Pre-RRS period (N=145) (%) | Post-RRS period (N=129) (%) | Total (N=274) (%) | P value |
|---|---|---|---|---|
| Age (years) | 72.00 (61.00–80.00) | 68.00 (58.00–74.00) | 69.00 (60.00–78.00) | 0.041 |
| Sex, female | 61 (42.36) | 63 (48.84) | 124 (45.42) | 0.342 |
| Type of Cancer | | | | |
| Others | 12 (8.28) | 12 (9.30) | 24 (8.76) | |
| Gynecologic | 27 (18.62) | 36 (27.91) | 63 (22.99) | |
| Hematology | 4 (2.76) | 4 (3.10) | 8 (2.92) | |
| Hepatobiliary | 17 (11.72) | 11 (8.53) | 28 (10.22) | |
| Lower gastrointestinal | 30 (20.69) | 33 (25.58) | 63 (22.99) | |
| Lung | 18 (12.41) | 12 (9.30) | 30 (10.95) | |
| Upper gastrointestinal | 37 (25.52) | 21 (16.28) | 58 (21.17) | |
| Stage | | | | < 0.001 |
| Localized | 79 (68.10) | 51 (43.97) | 130 (56.03) | |
| Metastatic | 37 (31.90) | 65 (56.03) | 102 (43.97) | |
| Site of Operation | | | | 0.190 |
| Others | 3 (2.07) | 1 (0.78) | 4 (1.46) | |
| Hepatobiliary | 12 (8.28) | 6 (4.65) | 18 (6.57) | |
| Lower gastrointestinal | 55 (37.93) | 50 (38.76) | 105 (38.32) | |
| Lung | 10 (6.90) | 3 (2.33) | 13 (4.74) | |
| Upper gastrointestinal | 65 (44.83) | 69 (53.49) | 134 (48.91) | |
| Cause of Operation | | | | 0.005 |
| Others | 8 (5.52) | 4 (3.10) | 12 (4.38) | |
| Bleeding | 25 (17.24) | 14 (10.85) | 39 (14.23) | |
| Ischemia | 12 (8.28) | 16 (12.40) | 28 (10.22) | |
| Obstruction | 26 (17.93) | 8 (6.20) | 34 (12.41) | |
| Perforation | 74 (51.03) | 87 (67.44) | 161 (58.76) | |
| ASA | | | | 0.124 |
| Low (<4) | 120 (82.76) | 96 (74.42) | 216 (78.83) | |
| High (≥4) | 25 (17.24) | 33 (25.58) | 58 (21.17) | |
| Hypertension | 51 (35.17) | 51 (39.84) | 102 (37.36) | 0.502 |
| Diabetes Mellitus | 37 (25.52) | 25 (19.53) | 62 (22.71) | 0.301 |
| Cardiovascular disease | 4 (2.76) | 5 (3.91) | 9 (3.30) | 0.738 |
| Chronic obstructive pulmonary disease | 6 (4.14) | 6 (4.65) | 12 (4.38) | 1.000 |
| Chronic kidney disease | 0 (0.0) | 4 (3.10) | 4 (1.46) | 0.048 |
| Liver disease | 4 (2.76) | 3 (2.33) | 7 (2.55) | 1.000 |
| APACHE II score | 23.00 (15.00–31.00) | 27.00 (20.00–32.00) | 25.00 (16.00–31.00) | 0.009 |
| SOFA score | | | | |
| Operation day | 4.00 (2.00–6.00) | 4.00 (3.00–8.00) | 4.00 (2.00–7.00) | 0.100 |
| 7 days after operation | 4.00 (2.00–6.00) | 3.00 (2.00–5.00) | 3.00 (2.00–6.00) | 0.007 |
| Delta SOFA score | 0.00 (-2.00–2.00) | -2.00 (-4.00–0.00) | -1.00 (-3.00–1.00) | 0.000 |
| Lactic Acid (mg/dL) | 34.70 (20.95–58.15) | 27.20 (13.90–53.00) | 30.55 (15.00–54.80) | 0.135 |
| Preoperative Fluid administration (L) | 0.30 (0.30–0.50) | 0.50 (0.30–1.30) | 0.30 (0.30–1.00) | 0.002 |
| ICU Admission | | | | 0.719 |
| Postoperative | 61 (42.07) | 58 (44.96) | 119 (43.43) | |
| Preoperative | 84 (57.93) | 71 (55.04) | 155 (56.57) | |
| Diagnosis to ICU admission (hour) | 8.77 (5.28–14.33) | 5.83 (1.92–9.36) | 7.12 (4.15–12.05) | < 0.001 |

*(Continued)*

**Table 1.** (Continued)

| | Pre-RRS period (N=145) (%) | Post-RRS period (N=129) (%) | Total (N=274) (%) | P value |
|---|---|---|---|---|
| Diagnosis to Operation (hour) | 6.48 (2.20–12.70) | 4.57 (2.98–7.22) | 5.20 (2.63–10.05) | 0.036 |
| Diagnosis to Antibiotics administration (hour) | 2.50 (1.13–6.92) | 2.01 (1.09–3.83) | 2.23 (1.12–5.00) | 0.042 |
| Vasopressor | | | | |
| Preoperative | 32 (22.07) | 47 (36.43) | 79 (28.83) | 0.013 |
| Postoperative | 72 (49.66) | 77 (59.69) | 149 (54.38) | 0.123 |
| 7 days after operation | 10 (7.75) | 5 (4.07) | 15 (5.95) | 0.332 |
| Duration (days) | 1.62 (0.49–3.45) | 1.50 (0.84–2.05) | 1.53 (0.67–2.58) | 0.584 |
| Mechanical ventilation | | | | |
| Preoperative | 10 (6.90) | 27 (20.93) | 37 (13.50) | 0.001 |
| Postoperative | 69 (47.59) | 78 (60.47) | 147 (53.65) | 0.044 |
| 7 days after operation | 13 (10.08) | 12 (9.76) | 25 (9.92) | 1.000 |
| Duration (days) | 3.00 (2.00–4.00) | 2.00 (2.00–4.00) | 2.00 (2.00–4.00) | 0.618 |
| CRRT | 6 (4.14) | 6 (4.65) | 12 (4.38) | 1.000 |
| Organ dysfunction | | | | 0.001 |
| Improve | 46 (35.66) | 61 (51.26) | 107 (43.15) | |
| Stability | 43 (33.33) | 44 (36.97) | 87 (35.08) | |
| Progression | 40 (31.01) | 14 (11.76) | 54 (21.77) | |
| ICU length of stay (days) | 1.78 (0.69–4.61) | 2.32 (0.93–4.61) | 1.91 (0.81–4.61) | 0.144 |
| Postoperative hospital stay (days) | 18.56 (11.82–37.12) | 22.53 (11.83–36.93) | 20.64 (11.82–37.12) | 0.364 |
| Survival discharge | | | | 0.039 |
| Death | 37 (25.52) | 19 (14.73) | 56 (20.44) | |
| Survival | 108 (74.48) | 110 (85.27) | 218 (79.56) | |

Values are presented as mean (SD), median (IQR), or n (%). Delta SOFA was defined as the SOFA score on postoperative day 7 minus the SOFA score on the operative day.

APACHE II, Acute Physiology and Chronic Health Evaluation II; ASA, American Society of Anesthesiologists; CRRT, continuous renal replacement therapy; ICU, Intensive Care Unit; RRS, rapid response system; SOFA, Sequential Organ Failure Assessment.

hours (N = 62). Baseline demographic and oncologic characteristics, including age, sex, cancer type, disease stage, and most comorbidities, were broadly comparable between groups. However, several clinical severity and perioperative variables differed: patients presenting outside coverage hours had higher APACHE II scores (28.00 vs. 25.00, p = 0.002), a greater proportion of postoperative ICU admissions (69.35% vs. 43.43%, p < 0.001), and longer intervals from diagnosis to operation (7.66 hours vs. 5.20 hours, p < 0.001). Postoperative mechanical ventilation duration was also longer in this group (4.50 days vs. 2.00 days, p = 0.003). Survival to discharge was lower among patients presenting outside coverage hours (64.52% vs. 79.56%, p = 0.018). Other perioperative parameters and postoperative outcomes showed no significant differences (S1 Table).

To further assess the potential clinical impact of limited RRS availability during these hours, an exploratory counterfactual analysis was conducted. Among patients who presented outside RRS coverage hours in the pre-RRS period, the observed survival rate was 59.4%. Using an adjusted logistic regression model that included RRS implementation, APACHE II score, SOFA score, lactate level, age, and cancer stage, the counterfactual predicted survival rate increased to 71.0% when RRS activation was hypothetically applied. This reflects an absolute survival difference of 11.6 percentage points. Detailed estimates are provided in S3 Table and a graphical summary is shown in S1 Fig.

## Comparison of preoperative and postoperative ICU admissions

The preoperative ICU admission group had a higher proportion of patients with ASA scores ≥4 (31.61% vs. 7.56%, p<0.001) and a higher incidence of cardiovascular disease (5.93% vs. 1.29%, p=0.043), along with elevated SOFA scores on the day of surgery (5.00 vs. 3.00, p<0.001) and on postoperative day 7 (4.00 vs. 2.00, p<0.001). Lactic acid levels were also higher in the preoperative ICU group (42.00 vs. 22.40 mg/dL, p<0.001). This group also received a larger volume of preoperative crystalloid fluids (0.50 L vs. 0.30 L, p<0.001). The time from diagnosis of acute abdomen to ICU admission was shorter in the preoperative group (5.50 vs. 8.95 h, p<0.001). Additionally, vasopressor use and mechanical ventilation were higher in the preoperative ICU admission group both before surgery (44.52% vs. 8.40%, p<0.001 and 23.23% vs. 0.84%, p<0.001) and after surgery (72.90% vs. 30.25%, p<0.001 and 73.55% vs. 27.73%, p<0.001), with higher use also observed on postoperative day 7 for both vasopressors (10.37% vs. 0.85%, p=0.004) and mechanical ventilation (17.78% vs. 0.85%, p<0.001). CRRT use was also more frequent in the preoperative group (7.74% vs. 0%, p=0.005). Patterns of organ dysfunction were similar between the preoperative and postoperative ICU admission groups, with improvement (45.52% vs. 40.35%), stability (32.09% vs. 38.60%), and progression (22.39% vs. 21.05%) (p=0.556). No significant differences were observed between the two groups regarding age, sex distribution, comorbidities (e.g., hypertension, diabetes), RRS implementation, or the time from diagnosis of acute abdomen to surgery and to antibiotic administration. Likewise, the groups did not differ significantly in the proportions of metastatic disease, the causes of surgery, or the indications for surgery. The ICU length of stay and postoperative hospital stay were significantly longer in the preoperative ICU admission group (ICU length of stay: 3.56 vs. 0.87 days, p<0.001; postoperative hospital stay: 23.75 vs. 17.19 days, p=0.002). In contrast, survival to discharge was higher in the postoperative group (93.28% vs. 69.03%, p<0.001; Table 2).

An interaction analysis was additionally performed to assess whether the effect of RRS implementation differed between patients requiring preoperative versus postoperative ICU admission. The interaction term was not statistically significant, indicating that RRS did not demonstrate a differential impact based on timing of ICU admission (S2 Table).

## Factors associated with survival: Univariable and multivariable logistic regression

Univariable logistic regression identified several variables associated with survival (Table 3). RRS implementation (OR, 1.983; 95% CI, 1.085–3.724; p=0.029), APACHE II score (OR, 0.949; 95% CI, 0.916–0.981; p=0.003), SOFA score (OR, 0.918; 95% CI, 0.856–0.984; p=0.015), and lactate level (OR, 0.984; 95% CI, 0.974–0.995; p=0.003) were associated with survival. Metastatic disease (OR, 0.390; 95% CI, 0.193–0.764; p=0.007) and preoperative ICU admission (OR, 0.161; 95% CI, 0.068–0.338; p<0.001) were associated with decreased survival. Vasopressor and mechanical ventilation use, both before and after surgery, as well as CRRT use, also showed significant associations in univariable analyses. Organ dysfunction category and study year were evaluated but did not demonstrate significant associations with survival.

In the multivariable analysis, RRS implementation remained independently associated with survival (adjusted OR, 3.554; 95% CI, 1.226–10.82; p=0.021). Metastatic disease (adjusted OR, 0.187; 95% CI, 0.060–0.509; p=0.002), preoperative ICU admission (adjusted OR, 0.202; 95% CI, 0.054–0.633; p=0.010), APACHE II score (adjusted OR, 0.927; 95% CI, 0.865–0.988; p=0.025), and CRRT use (adjusted OR, 0.183; 95% CI, 0.030–0.915; p=0.046) remained significant. Lactate level and organ dysfunction category were not retained in the final model, and study year was evaluated as a covariate to account for temporal improvement in institutional practice but was not independently associated with survival (Table 3) (Fig 2).

## Impact of RRS on survival: Kaplan-Meier analysis

Kaplan-Meier survival analysis was used to compare survival rates between the pre-RRS and post-RRS cohorts. As shown in Fig 3, the survival curve for the post-RRS group consistently remained higher than that of the pre-RRS group throughout the study period, with a statistically significant difference between the groups (log-rank test, p=0.039). At the

**Table 2. Comparison of pre- and postoperative ICU admissions of patients with acute abdomen.**

| | Preoperative ICU admission (N = 155) (%) | Postoperative ICU admission (N = 119) (%) | Total (N = 274) (%) | p-value |
|---|---|---|---|---|
| Age | 70.00 (60.00–79.00) | 69.00 (60.00–77.00) | 69.00 (60.00–78.00) | 0.463 |
| Sex, female | 74 (48.05) | 50 (42.02) | 124 (45.42) | 0.384 |
| RRS | | | | 0.719 |
| No | 84 (54.19) | 61 (51.26) | 145 (52.92) | |
| Yes | 71 (45.81) | 58 (48.74) | 129 (47.08) | |
| Type of Cancer | | | | 0.685 |
| Others | 15 (9.68) | 9 (7.56) | 24 (8.76) | |
| Gynecologic | 41 (26.45) | 22 (18.49) | 63 (22.99) | |
| Hematology | 4 (2.58) | 4 (3.36) | 8 (2.92) | |
| Hepatobiliary | 16 (10.32) | 12 (10.08) | 28 (10.22) | |
| Lower gastrointestinal | 31 (20.00) | 32 (26.89) | 63 (22.99) | |
| Lung | 16 (10.32) | 14 (11.76) | 30 (10.95) | |
| Upper gastrointestinal | 32 (20.65) | 26 (21.85) | 58 (21.17) | |
| Stage | | | | 0.837 |
| Localized | 73 (57.03) | 57 (54.81) | 130 (56.03) | |
| Metastatic | 55 (42.97) | 47 (45.19) | 102 (43.97) | |
| Site of operation | | | | 0.270 |
| Others | 3 (1.94) | 1 (0.84) | 4 (1.46) | |
| Hepatobiliary | 13 (8.39) | 5 (4.20) | 18 (6.57) | |
| Lower gastrointestinal | 63 (40.65) | 42 (35.29) | 105 (38.32) | |
| Lung | 5 (3.23) | 8 (6.72) | 13 (4.74) | |
| Upper gastrointestinal | 71 (45.81) | 63 (52.94) | 134 (48.91) | |
| Cause of Operation | | | | 0.229 |
| Others | 7 (4.52) | 5 (4.20) | 12 (4.38) | |
| Bleeding | 26 (16.77) | 13 (10.92) | 39 (14.23) | |
| Ischemia | 18 (11.61) | 10 (8.40) | 28 (10.22) | |
| Obstruction | 14 (9.03) | 20 (16.81) | 34 (12.41) | |
| Perforation | 90 (58.06) | 71 (59.66) | 161 (58.76) | |
| ASA | | | | < 0.001 |
| Low (<4) | 106 (68.39) | 110 (92.44) | 216 (78.83) | |
| High (≥4) | 49 (31.61) | 9 (7.56) | 58 (21.17) | |
| HTN | 52 (33.55) | 50 (42.37) | 102 (37.36) | 0.172 |
| Diabetes Mellitus | 39 (25.16) | 23 (19.49) | 62 (22.71) | 0.336 |
| Cardiovascular disease | 2 (1.29) | 7 (5.93) | 9 (3.30) | 0.043 |
| Chronic obstructive lung disease | 5 (3.23) | 7 (5.88) | 12 (4.38) | 0.443 |
| Chronic kidney disease | 3 (1.94) | 1 (0.84) | 4 (1.46) | 0.635 |
| Liver disease | 5 (3.23) | 2 (1.68) | 7 (2.55) | 0.703 |
| APACHE II | 26.00 (18.50–31.50) | 24.00 (16.00–30.00) | 25.00 (16.00–31.00) | 0.078 |
| SOFA | | | | |
| Operation day | 5.00 (3.00–8.00) | 3.00 (2.00–5.00) | 4.00 (2.00–7.00) | < 0.001 |
| 7 days after operation | 4.00 (3.00–6.00) | 2.00 (1.00–4.00) | 3.00 (2.00–6.00) | 0.000 |
| Delta SOFA score | −1.00 (−4.00–1.00) | −1.00 (−2.00–1.00) | −1.00 (−3.00–1.00) | 0.242 |
| Lactic acid (mg/dL) | 42.00 (19.00–69.30) | 22.40 (13.70–35.35) | 30.55 (15.00–54.80) | < 0.001 |
| Preoperative fluid administration (L) | 0.50 (0.30–1.36) | 0.30 (0.30–0.50) | 0.30 (0.30–1.00) | < 0.001 |

*(Continued)*

**Table 2.** (Continued)

| | Preoperative ICU admission (N = 155) (%) | Postoperative ICU admission (N = 119) (%) | Total (N = 274) (%) | *p-value* |
|---|---|---|---|---|
| Diagnosis to ICU admission (hour) | 5.50 (2.03–9.65) | 8.95 (6.35–13.13) | 7.12 (4.15–12.05) | < 0.001 |
| Diagnosis to operation (hour) | 4.85 (2.19–9.64) | 5.50 (3.23–10.21) | 5.20 (2.63–10.05) | 0.127 |
| Diagnosis to antibiotics administration (hour) | 2.12 (1.05–4.65) | 2.43 (1.13–5.43) | 2.23 (1.12–5.00) | 0.399 |
| Vasopressor | | | | |
| Preoperative | 69 (44.52) | 10 (8.40) | 79 (28.83) | < 0.001 |
| Postoperative | 113 (72.90) | 36 (30.25) | 149 (54.38) | < 0.001 |
| 7 days after operation | 14 (10.37) | 1 (0.85) | 15 (5.95) | 0.004 |
| Duration (days) | 1.71 (0.95–2.91) | 0.73 (0.42–1.76) | 1.53 (0.67–2.58) | 0.002 |
| Mechanical ventilator | | | | |
| Preoperative | 36 (23.23) | 1 (0.84) | 37 (13.50) | < 0.001 |
| Postoperative | 114 (73.55) | 33 (27.73) | 147 (53.65) | < 0.001 |
| 7 days after operation | 24 (17.78) | 1 (0.85) | 25 (9.92) | 0.000 |
| Duration (days) | 3.00 (2.00–5.00) | 2.00 (1.00–3.00) | 2.00 (2.00–4.00) | 0.022 |
| CRRT | 12 (7.74) | 0 (0.0) | 12 (4.38) | 0.005 |
| Organ dysfunction | | | | 0.556 |
| Improve | 61 (45.52) | 46 (40.35) | 107 (43.15) | |
| Stability | 43 (32.09) | 44 (38.60) | 87 (35.08) | |
| Progression | 30 (22.39) | 24 (21.05) | 54 (21.77) | |
| ICU length of stay (days) | 3.56 (1.79–7.04) | 0.87 (0.63–1.60) | 1.91 (0.81–4.61) | < 0.001 |
| Postoperative hospital stay (days) | 23.75 (12.79–45.86) | 17.19 (11.61–27.35) | 20.64 (11.82–37.12) | 0.002 |
| Survival Discharge | | | | < 0.001 |
| Death | 48 (30.97) | 8 (6.72) | 56 (20.44) | |
| Survival | 107 (69.03) | 111 (93.28) | 218 (79.56) | |

Values are presented as mean (SD), median (IQR), or n (%). Delta SOFA was defined as the SOFA score on postoperative day 7 minus the SOFA score on the operative day.

APACHE II, Acute Physiology and Chronic Health Evaluation II; ASA, American Society of Anesthesiologists; CRRT, continuous renal replacement therapy; ICU, intensive care unit; SOFA, Sequential Organ Failure Assessment.

30-day follow-up, the survival rate was 77.87% (95% CI: 68.72–84.63%) in the pre-RRS group and 86.70% (95% CI: 76.80–92.58%) in the post-RRS group. By the 60-day mark, survival in the pre-RRS group had declined to 55.86% (95% CI: 40.99–68.37%), while the post-RRS group maintained a higher survival rate of 75.38% (95% CI: 59.84–85.59%). Due to the limited number of events and the lack of 50% survival attainment in the post-RRS group, a median survival time could not be estimated for this group.

## Discussion

This study is the first to investigate the impact of the RRS on patients requiring emergency surgery for acute abdomen, providing valuable insights into the utility of the RRS for this high-risk group. The improved survival observed during the RRS period suggests that such systems can enhance patient outcomes by enabling timely interventions.

Despite the post-RRS cohort presenting with a higher proportion of metastatic cancer and overall greater physiologic severity, reflected by higher APACHE II scores and increased reliance on vasopressors and mechanical ventilation both before and after surgery, the time from diagnosis to ICU admission, surgical intervention, and antibiotic administration

**Table 3. Factors associated with survival of ICU patients with acute abdomen: univariable and multivariable logistic regression analysis.**

| | Univariable analysis | | Multivariable analysis | |
|---|---|---|---|---|
| | OR (95% CI) | P-value | OR (95% CI) | P-value |
| Sex (Female) | 0.911 (0.503–1.655) | 0.758 | | |
| Age (years) | 0.983 (0.959–1.007) | 0.165 | | |
| RRS | 1.983 (1.085–3.724) | 0.029 | 3.554 (1.226–10.82) | 0.021 |
| Type of Cancer | | | | |
| Others | Reference | | | |
| Gynecologic | 1.167 (0.367–3.406) | 0.783 | | |
| Hematology | 1.000 (0.170–8.075) | >0.999 | | |
| Hepatobiliary | 1.533 (0.400–6.109) | 0.531 | | |
| Lower gastrointestinal | 1.282 (0.401–3.786) | 0.660 | | |
| Lung | 1.095 (0.304–3.867) | 0.887 | | |
| Upper gastrointestinal | 1.815 (0.542–5.783) | 0.316 | | |
| Metastatic | 0.390 (0.193–0.764) | 0.007 | 0.187 (0.060–0.509) | 0.002 |
| Site of Operation | | | | |
| Others | Reference | | | |
| Hepatobiliary | 2.667 (0.105–38.88) | 0.476 | | |
| Lower gastrointestinal | 0.833 (0.040–6.803) | 0.877 | | |
| Lung | 1.833 (0.071–27.22) | 0.662 | | |
| Upper gastrointestinal | 1.794 (0.086–14.79) | 0.620 | | |
| Cause of operation | | | | |
| Others | Reference | | | |
| Bleeding | 2.400 (0.287–16.51) | 0.372 | | |
| Ischemia | 0.920 (0.118–5.118) | 0.928 | | |
| Obstruction | 0.650 (0.088–3.178) | 0.622 | | |
| Perforation | 0.647 (0.097–2.593) | 0.585 | | |
| ASA | | | | |
| Low (<4) | Reference | | | |
| High (≥4) | 0.531 (0.276–1.048) | 0.062 | | |
| APACHE II | 0.949 (0.916–0.981) | 0.003 | 0.927 (0.865–0.988) | 0.025 |
| SOFA<br>Delta SOFA score | 0.918 (0.856–0.984)<br>0.940 (0.844–1.041) | 0.015<br>0.247 | | |
| Lactic acid (mg/dL) | 0.984 (0.974–0.995) | 0.003 | 1.00 (0.984–1.015) | 0.945 |
| Preoperative fluid administration (L) | 0.941 (0.657–1.405) | 0.751 | | |
| ICU admission | | | | |
| Postoperative | Reference | | | |
| Preoperative | 0.161 (0.068–0.338) | <0.001 | 0.202 (0.054–0.633) | 0.010 |
| Diagnosis to ICU admission (hour) | 1.027 (0.981–1.080) | 0.276 | | |
| Diagnosis to operation (hour) | 0.973 (0.927–1.024) | 0.289 | | |
| Diagnosis to antibiotics administration (hour) | 1.012 (0.936–1.105) | 0.783 | | |
| Preoperative vasopressor use | 0.336 (0.182–0.620) | <0.001 | | |
| Postoperative vasopressor use | 0.223 (0.105–0.439) | <0.001 | | |
| Duration of Vasopressor (days) | 0.887 (0.790–0.959) | 0.015 | | |
| Preoperative mechanical ventilation | 0.409 (0.195–0.887) | 0.020 | | |
| Postoperative mechanical ventilation | 0.310 (0.155–0.587) | <0.001 | | |
| Duration of mechanical ventilator (days) | 0.951 (0.899–1.005) | 0.075 | | |
| CRRT | 0.112 (0.029–0.371) | <0.001 | 0.183 (0.030–0.915) | 0.046 |

*(Continued)*

**Table 3.** (Continued)

| | Univariable analysis | | Multivariable analysis | |
|---|---|---|---|---|
| | OR (95% CI) | P-value | OR (95% CI) | P-value |
| Organ Failure | | | | |
| Improve | Reference | | | |
| Stability | 0.789 (0.333–1.873) | 0.588 | | |
| Progression | 0.494 (0.201–1.222) | 0.122 | | |
| Study year | 1.025 (0.888–1.185) | 0.736 | 1.274 (0.945–1.755) | 0.122 |

Delta SOFA was defined as the SOFA score on postoperative day 7 minus the SOFA score on the operative day.

APACHE II, Acute Physiology and Chronic Health Evaluation II; ASA, American Society of Anesthesiologists; CRRT, continuous renal replacement therapy; ICU, Intensive Care Unit; RRS, rapid response system; SOFA, Sequential Organ Failure Assessment.

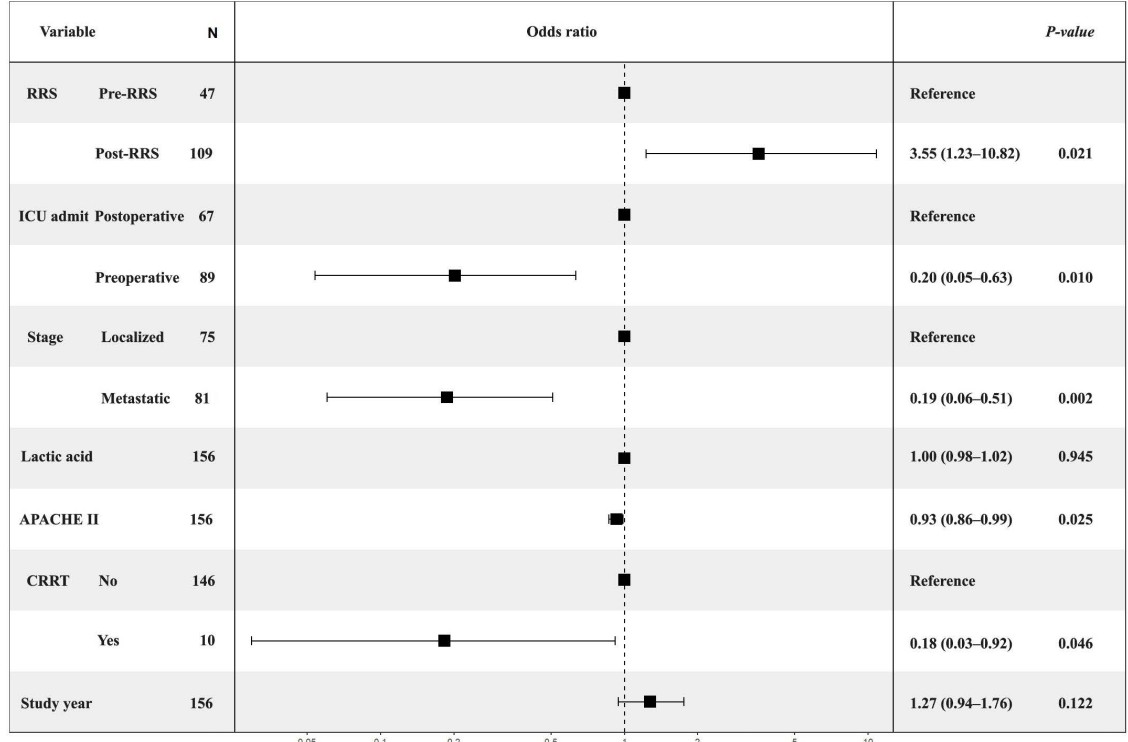

**Fig 2. Forest plot of odds ratios for prognostic factors in ICU patients with acute abdomen (March 2016–August 2023).** APACHE, Acute Physiology and Chronic Health Evaluation; CRRT, Continuous Renal Replacement Therapy; ICU, Intensive Care Unit; RRS, Rapid Response System. This figure presents the odds ratios and 95% confidence intervals for variables included in the multivariable logistic regression model. The post-RRS period, preoperative ICU admission, metastatic disease, higher APACHE II score, and CRRT use were independently associated with increased in-hospital mortality, whereas lactic acid level and study year were not significant predictors.

was shorter following RRS implementation. Beyond improvements in timeliness, patients in the post-RRS period also demonstrated more favorable patterns of organ dysfunction. SOFA scores on the day of operation were similar between groups, but postoperative day 7 SOFA scores were lower in the RRS group, and the reduction in SOFA from baseline was greater, indicating that progression of organ dysfunction was mitigated after RRS activation. Consistent with this trend, fewer patients required vasopressor or ventilator support one week postoperatively, suggesting attenuation of physiologic

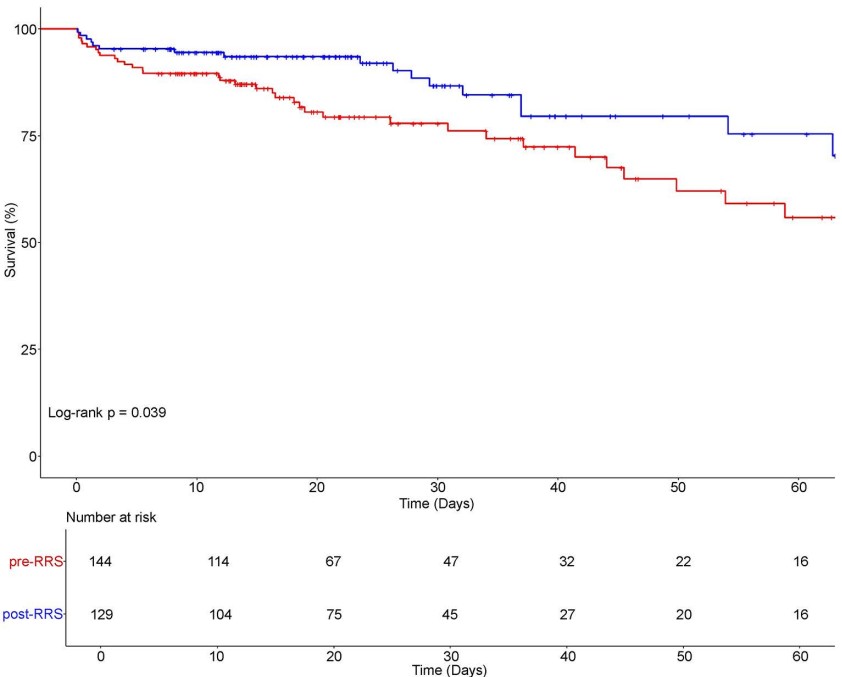

**Fig 3. Kaplan–Meier survival curve for ICU patients with acute abdomen, comparing RRS implementation (March 2016–August 2023).** ICU, intensive care unit; RRS, Rapid Response System; The curve shows the survival probability of patients over time (in days) based on the presence or absence of the RRS. The blue line represents the period when the RRS was operational, while the red line indicates the period without RRS intervention. The number-at-risk table below the plot displays the number of patients remaining under observation at different time points.

deterioration. Although preoperative vasopressor and ventilator use was higher in the post-RRS group, the subsequent increase in postoperative organ support was not greater than that observed in the pre-RRS period, suggesting that RRS involvement may have helped stabilize patients before surgery. Collectively, these findings indicate that the benefits observed during the RRS period were not limited to earlier interventions but extended to improved postoperative physiologic trajectories and decreased organ support requirements. Further comparison of patients who presented during outside coverage hours showed substantially worse baseline severity, longer delays to operative management, and greater postoperative organ support needs, resulting in lower survival compared with patients evaluated during coverage hours. In an exploratory counterfactual analysis, predicted survival among patients presenting outside coverage hours increased from 59.4% to 71.0% when RRS activation was hypothetically applied, suggesting the potential clinical value of extending RRS availability beyond current operational periods. These findings align with previous research indicating that early ICU admission and surgical intervention are crucial for improving outcomes in patients with acute abdomen [3,21–23].

Metastatic disease was a significant predictor of poor survival in this study, consistent with other research that links advanced-stage disease to worse outcomes following emergency surgery [24,25]. Even with rapid intervention, metastatic disease remains a challenging prognostic factor. Early ICU admission, as seen in the post-RRS period, provided a better perioperative management framework, yet these patients exhibited poorer outcomes due to advanced disease. These findings underscore that while timely interventions benefit patients with acute abdomen, metastatic disease remains a challenging prognostic factor, indicating the need for targeted perioperative strategies in managing patients with advanced cancer.

In our study, patients requiring ICU admission before surgery demonstrated substantially higher illness severity and worse postoperative outcomes than those transferred after the operation. This subgroup presented with elevated ASA/

SOFA scores, higher lactate levels, and greater dependence on vasopressors and mechanical ventilation, indicating significant physiological compromise at the time of diagnosis. The shorter time intervals to ICU admission and surgery observed after RRS implementation suggest that the system supported faster recognition and escalation of care for hemodynamically unstable patients. However, even with timely evaluation and transfer, the profound baseline instability in those requiring preoperative ICU admission remained a dominant factor influencing their postoperative prognosis. These findings are consistent with previous reports showing that early severe physiological derangement is strongly associated with adverse outcomes in high-risk surgical populations [26–27]. In the subgroup interaction analysis, RRS implementation did not show a differential survival effect between preoperative and postoperative ICU admissions, suggesting that the beneficial impact of RRS was consistent across both levels of preoperative stability. Collectively, these results highlight the importance of early identification and stabilization strategies for patients presenting with severe preoperative instability, and future work should aim to refine approaches to improve outcomes in this vulnerable group.

ICU patients who required vasopressor or ventilator support also had worse outcomes. Although these interventions are essential for stabilizing physiologically unstable patients, their use often reflects severe underlying illness rather than functioning as independent determinants of mortality [26–29]. Similarly, elevated lactate levels were associated with survival in the univariable analyses, but their prognostic effect diminished after adjustment for composite severity measures. Vasopressor use, mechanical ventilation, and lactate therefore appear to function primarily as markers of global physiologic derangement, which is more comprehensively captured by variables such as metastatic disease status and APACHE II score. This interpretation is consistent with previous studies showing that once overall severity is considered, individual markers of instability lose their predictive independence [30,31]. These findings highlight the importance of accurate assessment of initial physiologic severity and timely escalation of care in high-risk patients. In this context, the RRS may facilitate earlier recognition of clinical deterioration and support timely intervention and optimized perioperative management.

CRRT use was associated with worse survival, likely because patients requiring CRRT often have multiorgan failure, which exacerbates disease severity. This finding is consistent with existing literature indicating that the need for renal replacement therapy in critically ill patients signals a higher risk of mortality [32,33].

We also found that higher APACHE II scores were associated with poorer survival. This is consistent with the use of the APACHE II score as a predictor of mortality in critically ill patients, reflecting disease severity. In patients with cancer presenting with acute abdomen, elevated APACHE II scores likely capture the cumulative burden of systemic instability at presentation, which limits the degree to which rapid interventions can alter postoperative outcomes. These findings align with previous studies demonstrating the importance of initial disease severity in determining prognosis [34,35].

This study has several limitations. First, the single-center nature, retrospective design, and modest sample size may limit the generalizability of the findings. The cancer cohort also included patients with a wide range of clinical conditions, and this heterogeneity could not be fully characterized using only a localized versus metastatic grouping. Important oncologic information such as recent chemotherapy, immunotherapy, treatment response, and performance status was not available, which may have influenced baseline severity and postoperative outcomes. Second, the RRS operated only from 6:00 AM to 11:00 PM on weekdays, resulting in the exclusion of patients who deteriorated outside coverage hours. Because nighttime deterioration often occurs during periods of reduced staffing and delayed recognition, the absence of these cases may have led to underestimation of the full potential impact of the RRS. Nevertheless, our supplemental analysis showed that patients presenting during and outside coverage hours were largely comparable in most demographic and oncologic characteristics, although residual selection bias cannot be entirely excluded. Third, the use of convenience sampling may have introduced selection bias, despite efforts to reduce this risk through supplemental comparisons. Additionally, although we examined the interaction between RRS implementation and preoperative ICU admission, the study was not powered to detect subgroup-specific effects, and these results should be interpreted cautiously. Furthermore, the counterfactual analysis used to estimate the potential benefit of extending RRS availability to outside coverage hours was

exploratory and model-based; its findings rely on assumptions inherent to regression modeling and should not be interpreted as establishing causal effects. Finally, although this study demonstrated improved outcomes following RRS implementation, detailed information on the specific types of RRS interventions was not consistently available, making it difficult to determine which elements of the RRS contributed most to the observed effect. Future studies with larger cohorts across multiple tertiary cancer centers will be important to verify whether the observed effects are consistent in different clinical environments. Evaluating continuous, 24-hour RRS operation and integrating standardized, intervention-level documentation will help clarify which components most contribute to improved prognosis.

## Conclusions

Improved survival after RRS implementation appears to be associated with more timely perioperative care and attenuated postoperative organ dysfunction, suggesting that RRS activation may help stabilize high-risk surgical patients. Outcomes were also poorer during outside-coverage hours, and exploratory counterfactual estimates indicated potential benefit from extended availability. These findings support broader implementation and evaluation of continuous RRS coverage in future multicenter studies.

## Supporting information

**S1 Table. Comparison of clinical characteristics between patients admitted during hours with Rapid Response System coverage and those admitted outside coverage hours.** This table presents demographic factors, cancer-related characteristics, illness severity markers, perioperative time intervals, and postoperative outcomes in patients undergoing emergency surgery for acute abdomen. Variables compared include age, sex, cancer type and stage, APACHE II and SOFA scores, lactate level, organ support use, and survival to discharge. Differences between the groups were assessed using appropriate statistical tests, with results reported as mean (SD), median (IQR), or n (%).
(DOCX)

**S2 Table. Interaction Analysis of RRS Implementation and Preoperative ICU Admission on Survival.** Multivariable logistic regression was performed including RRS implementation, ICU admission timing, metastatic disease, APACHE II score, lactic acid level, CRRT use, and study year. The interaction term (RRS × preoperative ICU admission) was not statistically significant, indicating no evidence of a differential effect of RRS on survival based on timing of ICU admission. Adjusted odds ratios (ORs) with 95% confidence intervals (CIs) are presented. APACHE II, Acute Physiology and Chronic Health Evaluation II; CRRT, continuous renal replacement therapy; ICU, Intensive Care Unit; RRS, rapid response system.
(DOCX)

**S3 Table. Observed and Counterfactual Survival During Outside Coverage Hours.** Observed survival reflects the actual survival rate among patients who deteriorated outside coverage hours in the pre-RRS period. Counterfactual survival represents the predicted probability of survival if these patients had access to RRS activation, based on an adjusted logistic regression model incorporating RRS implementation, coverage status, APACHE II score, SOFA score, lactate level, age, and cancer stage. The absolute difference demonstrates the estimated magnitude of survival improvement under hypothetical RRS availability during these hours.
(DOCX)

**S1 Fig. Observed vs Counterfactual Survival During Outside Coverage Hours.** The left bar shows the observed survival probability (59.4%) among patients who deteriorated outside RRS coverage hours. The right bar displays the counterfactual survival probability (71.0%) predicted under a hypothetical scenario in which full RRS activation was available during these hours. Counterfactual estimates were generated using an adjusted logistic regression model including RRS

implementation, coverage status, APACHE II score, SOFA score, lactate level, age, and cancer stage. The difference illustrates the potential survival benefit associated with extending RRS availability to periods currently without operational coverage.
(TIF)

## Author contributions

**Conceptualization:** Jae Hoon Lee, Ki Ho Yu, Sang Yun Jung.

**Data curation:** Jae Hoon Lee, Sang Yun Jung.

**Formal analysis:** Jae Hoon Lee, Won Ho Han.

**Investigation:** Mee Hee Shin, Yun Jung Choi, Ra Mi Choi, Woo Jin Seo, Sang Hee Park.

**Methodology:** Jae Hoon Lee, Ki Ho Yu.

**Writing – original draft:** Jae Hoon Lee, Ki Ho Yu.

**Writing – review & editing:** Jae Hoon Lee, Won Ho Han, Sang Yun Jung.

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
