## [Decision Letter · Decision Letter 0]

3 Nov 2025

Dear Dr. Han,

Thank you for submitting your manuscript to PLOS ONE. After careful consideration, we feel that it has merit but does not fully meet PLOS ONE’s publication criteria as it currently stands. Therefore, we invite you to submit a revised version of the manuscript that addresses the points raised during the review process.

We look forward to receiving your revised manuscript.

Kind regards,

Stefano Turi

Academic Editor

PLOS ONE

Journal Requirements:

3. We notice that your supplementary figures are uploaded with the file type 'Other'. Please amend the file type to 'Supporting Information'. Please ensure that each Supporting Information file has a legend listed in the manuscript after the references list.

Reviewers' comments:

Reviewer's Responses to Questions

**Comments to the Author**

1. Is the manuscript technically sound, and do the data support the conclusions?

Reviewer #1: Partly

Reviewer #2: Partly

2. Has the statistical analysis been performed appropriately and rigorously?

Reviewer #1: No

Reviewer #2: Yes

3. Have the authors made all data underlying the findings in their manuscript fully available?

Reviewer #1: Yes

Reviewer #2: Yes

4. Is the manuscript presented in an intelligible fashion and written in standard English?

Reviewer #1: Yes

Reviewer #2: No

Reviewer #1: The authors present a significant single-center study suggesting that the implementation of a Rapid Response System (RRS) improves survival among high-risk cancer patients undergoing emergency surgery for acute abdomen. However, several critical limitations concerning patient heterogeneity, clinical interpretation, and the RRS operational window must be addressed.

My main concerns are as follows:

1. The primary limitation of this study is the substantial clinical heterogeneity present within the cancer cohort, which is only loosely characterized by "Localized" versus "Metastatic" stage. This grouping obscures crucial prognostic differences, as the study lacks essential detail on active oncologic treatment—such as recent chemotherapy, immunotherapy, or performance status—all of which drastically affect emergency surgical outcomes. The broad spectrum of risk factors encompassed by this single-center cohort, albeit of modest size, imposes limitations on the generalizability of the findings and hinders the capacity to discern the RRS's impact on particular patient subgroups. In order to enhance the study's power and clinical relevance, the authors must discuss the necessity and feasibility of expanding the cohort size. This could be achieved by extending the observation period or through a multi-center collaboration.

2. The finding that preoperative ICU admission is an independent predictor of poorer survival (adjusted OR: 0.154, p<0.001) initially appears to contradict the RRS goal of early intervention. However, the data demonstrate that patients admitted preoperatively were the most critically ill, exhibiting significantly higher APACHE II, SOFA, and lactate scores, and greater reliance on preoperative vasopressors and mechanical ventilation (all p<0.001 in secondary analysis). Consequently, preoperative ICU admission should not be interpreted as a failure of timely care, but rather as an indicator of overwhelming, severe physiological derangement present at the time of diagnosis. The RRS effectively facilitated the rapid transfer of these patients, who were highly unstable. However, the patients' severe underlying condition dictated a poor prognosis, which is a crucial distinction that must be explicitly clarified in the discussion.

3. The RRS was operational only from 6:00 AM to 11:00 PM on weekdays. This limited insurance coverage resulted in the exclusion of 62 patients whose emergencies occurred during the nocturnal hours (11:00 p.m. – 6:00 a.m.). This is a significant methodological limitation because the RRS is typically expected to yield its greatest relative benefit during these off-hours, when staff-to-patient ratios and senior physician availability are lowest. The exclusion of this vulnerable time window precludes the study from drawing conclusions on the RRS's effectiveness in mitigating the "night-time effect." Consequently, the study may have underestimated the full potential impact of the RRS. It is imperative that the authors underscore this limitation and propose that subsequent research be conducted to substantiate the purported advantages of uninterrupted, around-the-clock RRS coverage.

Reviewer #2: Reviewer Comments

Question 1

It is verified that there is no logical contradiction between the two pieces of information in the manuscript PONED2516230_reviewer.pdf, but the temporal relationship needs to be clarified.

The study included 274 patients who underwent emergency surgery for acute abdomen between March 2016 and August 2023; this time period represents the patient enrollment and data collection phase of the retrospective cohort study, which is consistent with the retrospective nature of the study .

The data access period for the manuscript is specified as October 14, 2024, to December 14, 2024, which corresponds to the poststudy data verification and manuscript preparation phase. Since the study was approved by the Institutional Review Board (IRB) of the National Cancer Center with approval number NCC 20240268, the data access period is set after IRB approval to ensure compliance with ethical requirements for retrospective data use .

Additionally, the manuscript correctly states that "the requirement for informed consent from patients was waived owing to the retrospective nature of this study," which aligns with common ethical practices for retrospective cohort studies (i.e., waiving informed consent when historical data are used and patient privacy is protected) .

Question 2: Sentence Structure Errors

1. Tense Inconsistency:

Original sentence (Ethics Statement section): "This study was conducted in accordance with... and was approved... The data access period for the manuscript is from October 14, 2024... to December 14, 2024"

Issue: The first half uses the past tense ("was conducted," "was approved") to describe completed study procedures, while the second half incorrectly uses the present tense ("is") for the data access period. Since the data access period is a prespecified window approved by the IRB (and the manuscript is in the submission stage), the past tense should be used to maintain consistency.

Revised sentence: "This study was conducted in accordance with... and was approved... The data access period for the manuscript was from October 14, 2024... to December 14, 2024" .

2. Missing Logical Conjunctions:

Original sentences (Abstract section): "Kaplan–Meier analysis confirmed better survival over time in the postRRS group (logrank p = 0.039). Time intervals from diagnosis to critical interventions... were significantly shorter following RRS implementation"

Issue: The two sentences describe complementary findings of the study (improved survival and shortened intervention time), but lack a conjunction to clarify their parallel or causal relationship, leading to disjointed logic.

Revised sentences: "Kaplan–Meier analysis confirmed better survival over time in the postRRS group (logrank p = 0.039). Meanwhile, time intervals from diagnosis to critical interventions... were significantly shorter following RRS implementation"

(The conjunction "Meanwhile" emphasizes the parallelism between the two key results, strengthening logical coherence) .

Question 3: Insufficient Innovation

1. Limitations of Study Design Restricting Innovative Generalization:

The study adopts a singlecenter retrospective cohort design. Although the sample size (274 patients) was shown to have sufficient statistical power via posthoc analysis (power = 1.0 for Cohen’s d = 0.8), singlecenter data cannot rule out confounding effects from "institutionspecific clinical protocols" (e.g., concurrent improvements in anesthetic techniques or postoperative monitoring protocols). This limits the external validity of the innovative conclusions.

Recommendation:

Supplement a "comparative analysis of baseline characteristics between cases during RRS operational hours and nonoperational hours" to confirm that excluded cases (11:00 PM–6:00 AM) do not introduce selection bias .

Propose a "multicenter data validation plan" (e.g., collaborating with other tertiary cancer centers to replicate the study) to verify whether RRS exerts consistent effects across different institutions, thereby enhancing the generalizability of the innovative findings.

2. Insufficient Depth of Innovation:

The study does not explore the "specific effective components of RRS"—for example, which subcomponents of the RRS team (e.g., "intensivistled bedside assessment," "realtime vital sign monitoring by nurses," or "multidisciplinary rapid consultation") contribute more to improved prognosis. Without stratified or subgroup analyses to decompose these components, the innovative conclusion remains at the macro level of "RRS is effective," lacking detailed guidance for clinical practice.

Recommendation:

Add "subgroup analyses of the association between different RRS interventions (e.g., bedside stabilization only, ICU transfer, or emergency surgery coordination) and prognosis" to identify the most impactful components of RRS. This will elevate the clinical guiding value of the innovative findings by providing actionable insights for optimizing RRS implementation .

Question 4: Incomplete Logical Closure in Study Design

Issue 1: Uncontrolled Confounding from RRS Operational Hours

The manuscript explicitly states that RRS operates only "on weekdays from 6:00 AM to 11:00 PM" and excludes cases occurring between "11:00 PM and 6:00 AM." However, it fails to:

Analyze "baseline differences (e.g., disease severity, intervention measures) between excluded cases (nonoperational hours) and included cases (operational hours)";

Explain whether "cases during RRS operational hours included patients whose condition deteriorated at night but were delayed until daytime for RRS activation."

These omissions may introduce "mild disease bias" in the included sample (e.g., excluding more severely ill patients who deteriorate at night), weakening the causal inference that "RRS improves prognosis" .

Issue 2: Inadequate Explanation of Baseline Imbalance

Results show that the postRRS group had:

A higher proportion of metastatic disease (56.03% vs. 31.90%, p < 0.001);

Higher APACHE II scores (27 vs. 23, p = 0.009);

yet a significantly higher survival rate (85.27% vs. 74.48%, p = 0.039). The authors only attribute this to "timely intervention by RRS" but fail to rule out confounding factors from "concurrent improvements in hospitalwide clinical practices" (e.g., advances in emergency surgical techniques, optimized postoperative infection control, or enhanced ICU monitoring) between 2016–2019 (preRRS) and 2019–2023 (postRRS), leading to an incomplete causal chain .

Recommendations

1. Supplement a "table comparing baseline characteristics of cases during RRS operational vs. nonoperational hours" to justify the rationality of excluding nonoperational hour cases (e.g., confirming no significant differences in disease severity between the two groups) .

2. Use "propensity score matching (PSM)" to balance baseline variables (e.g., metastatic disease status, APACHE II scores) between the preRRS and postRRS groups; alternatively, include "study year" as a confounding variable in the multivariable regression model to explicitly rule out the impact of concurrent improvements in medical standards .

Question 5: Unsubstantiated Variable Selection Criteria

The manuscript mentions that "variables with p < 0.2 in univariable analysis were included in the multivariable regression" but provides no justification for this criterion (e.g., whether it is based on prior studies in the field or preanalysis validation). Additionally, it fails to report the reasons for excluding variables such as "SOFA score and lactate level" from the multivariable model. This may lead to the omission of key confounders, undermining the reliability of the results .

Recommendation:

In the "Statistical Analysis" section, supplement:

1. Literature support for the p < 0.2 criterion (e.g., citing methodological studies that recommend this threshold for variable selection in observational studies);

2. Preanalysis results (e.g., correlation analysis between variables) to explain why variables like SOFA score and lactate level were excluded (e.g., high collinearity with APACHE II score) .

Question 6: Lack of Interaction Analysis

The results show that "preoperative ICU admission was significantly associated with poor prognosis (adjusted OR = 0.154, p < 0.001)." However, the study does not analyze "whether RRS improves the prognosis of patients with preoperative ICU admission" (i.e., the interaction between RRS and preoperative ICU admission). This gap prevents clarification of RRS’s intervention value for the "most severely ill subgroup," limiting the depth of logical analysis .

Recommendation:

1. Add an interaction term "RRS × preoperative ICU admission" to the multivariable logistic regression model;

2. Report the adjusted OR and pvalue of the interaction term to determine whether RRS exerts differential effects on patients with preoperative vs. postoperative ICU admission. For example, if the interaction term is significant, it could indicate that RRS is more effective for patients with preoperative ICU admission (a severely ill subgroup) .

Question 7: Incomplete "InterventionPrognosis" Causal Chain

The manuscript claims that "RRS shortens key intervention time windows → improves survival" but fails to explain why shortened time windows lead to better prognosis. For example:

Whether shorter antibiotic administration time reduces sepsis incidence;

Whether shorter ICU admission time decreases the risk of multiple organ failure.

The lack of "intermediate outcome measures" (e.g., sepsis, organ failure) creates a break in the causal chain between "intervention measures" and "prognosis improvement" .

Recommendation:

Supplement "group comparisons of intermediate outcome measures," such as:

Incidence of sepsis (30day postsurgery);

Incidence of multiple organ failure (within 7 days of ICU admission);

This will complete the causal chain: "RRS → shortened time windows → reduced complications → improved survival," strengthening the logical rigor of the conclusion .

Question 8: Unexplained Contradictory Results

Baseline data show that the postRRS group had higher rates of:

Preoperative vasopressor use (36.43% vs. 22.07%, p = 0.013);

Preoperative mechanical ventilation (20.93% vs. 6.90%, p = 0.001);

(These indicators suggest more severe preoperative illness.) However, there were no significant differences in postoperative vasopressor or mechanical ventilation rates between the two groups. The authors do not interpret "whether RRS reduces postoperative organ support needs by stabilizing preoperative condition," resulting in fragmented result interpretation .

Recommendation:

1. Calculate "the difference in organ support needs (preoperative vs. postoperative)" for each group (e.g., "postoperative vasopressor use rate minus preoperative rate");

2. Compare these differences between the preRRS and postRRS groups to clarify whether RRS mitigates the progression of illness (e.g., a smaller increase in postoperative organ support needs in the postRRS group would indicate RRS’s stabilizing effect) .

Question 9: Spelling Errors

1. "climinating variables sequentially" (Statistical Analysis section):

Error: "climinating" (extra "c");

Correction: "eliminating" (meaning "to remove variables stepwise") .

2. "intation of antibiotic administration" (Baseline Characteristics section):

Error: "intation" (incorrect term; "intation" refers to "endotracheal intubation," which is unrelated to antibiotic administration);

Correction: "initiation" (meaning "the start of antibiotic administration") .

3. "APACHI II" (Forest Plot variable label):

Error: Missing "E" (abbreviation error);

Correction: "APACHE II" (full name: Acute Physiology and Chronic Health Evaluation II, a standard severityofillness score) .

Question 10: Punctuation and Formatting Errors

1. Duplicate pvalue Labeling:

Original text (Baseline Characteristics section): "the time from diagnosis of acute abdomen to ICU admission (5.83 vs. 8.77 h, p < 0.001), \(p<0.001\) ), surgery (4.57 vs.6.48 h, \(p=0.036\) )"

Error: Duplicate pvalue label ("p < 0.001") for ICU admission time; missing space between "vs." and "6.48";

Correction: "the time from diagnosis of acute abdomen to ICU admission (5.83 vs. 8.77 h, p < 0.001), surgery (4.57 vs. 6.48 h, p = 0.036)" .

2. Missing Subject:

Original text (Baseline Characteristics section): "Vasopressor use and mechanical ventilation were higher in the postRRS group preoperatively (36.43% vs.22.07%, \(p=0.013\) and 20.93% vs.6.90%, \(p=0.001\) , respectively)"

Errors: Missing subject ("were higher" incorrectly refers to "vasopressor use/mechanical ventilation" instead of their "rates"); misspelling ("prcoperatively" → "preoperatively"); missing spaces around "vs.";

Correction: "The rates of vasopressor use and mechanical ventilation were higher in the postRRS group preoperatively (36.43% vs. 22.07%, p = 0.013 and 20.93% vs. 6.90%, p = 0.001, respectively)" .

3. Disorganized Table Formatting:

In Table 1, "Preoperative Fluid Administration (mL)" row:

Inconsistent number formatting: "1300.00" (postRRS group) lacks a thousands separator (should be "1,300.00");

Cell content overflow: Values (e.g., "300.00 (300.00–1,000.00)") exceed cell width, reducing readability.

Recommendations:

Standardize number formatting (add thousands separators for numbers ≥1,000);

Adjust column widths to ensure values fit within cells without overflow .

**Do you want your identity to be public for this peer review?** For information about this choice, including consent withdrawal, please see our Privacy Policy

Reviewer #1: **Yes:**  Tomoyuki Abe

Reviewer #2: No

---

## [Author Response · Author response to Decision Letter 1]

5 Dec 2025

We respectfully submit a complete point-by-point response to all editorial and reviewer comments in the document entitled “Response to Reviewers.” All revisions requested have been addressed in the manuscript, and tracked changes have been included as required.

---

## [Decision Letter · Decision Letter 1]

11 Jan 2026

Evaluating the Impact of a Rapid Response System on Survival of Patients with Cancer Undergoing Emergency Surgery for Acute Abdomen: A Single-Center Retrospective Cohort Study

PONE-D-25-16230R1

Dear Dr. Ho Han,

We’re pleased to inform you that your manuscript has been judged scientifically suitable for publication and will be formally accepted for publication once it meets all outstanding technical requirements.

Kind regards,

Stefano Turi

Academic Editor

PLOS One

Reviewers' comments:

Reviewer's Responses to Questions

**Comments to the Author**

Reviewer #1: All comments have been addressed

2. Is the manuscript technically sound, and do the data support the conclusions?

Reviewer #1: Yes

3. Has the statistical analysis been performed appropriately and rigorously?

Reviewer #1: Yes

4. Have the authors made all data underlying the findings in their manuscript fully available?

Reviewer #1: Yes

5. Is the manuscript presented in an intelligible fashion and written in standard English?

Reviewer #1: Yes

Reviewer #1: The authors have addressed all reviewer concerns with high academic rigor, particularly through the inclusion of a counterfactual model for nocturnal RRS coverage and serial SOFA trajectory analyses to establish a mechanistic link between timely intervention and improved outcomes. By reframing preoperative ICU admission as a marker of disease severity and adjusting for study year to account for secular trends, they have significantly strengthened the logical coherence of their findings. The revisions to the limitations regarding cancer heterogeneity further enhance the transparency of the study. Consequently, the manuscript now provides robust clinical insights and is recommended for publication in its current form.

**Do you want your identity to be public for this peer review?** For information about this choice, including consent withdrawal, please see our Privacy Policy

Reviewer #1: **Yes:**  Tomoyuki Abe

---

## [Editor Report · Acceptance letter]

PONE-D-25-16230R1

PLOS One

Dear Dr. Han,

I'm pleased to inform you that your manuscript has been deemed suitable for publication in PLOS One. Congratulations! Your manuscript is now being handed over to our production team.

Kind regards,

on behalf of

Dr. Stefano Turi

Academic Editor

PLOS One